# Engineering of a high-fidelity Cas12a nuclease variant capable of allele-specific editing

**Jingjing Wei**[1,2], **Jingtong Liu**[2], **Ziwen Wang**[3], **Yuan Yang**[2], **Yuwen Tian**[2], **Shengzhou Wang**[2], **Bao-Qing Gao**[4,5,6], **Song Gao**[3], **Li Yang**[5,6], **Junnan Tang**[1]\*, **Yongming Wang** [1,2]\*

1 Department of Cardiology, the First Affiliated Hospital of Zhengzhou University, Zhengzhou, China,
2 Center for Medical Research and Innovation, Shanghai Pudong Hospital, Fudan University Pudong Medical Center, School of Life Sciences, Shanghai Engineering Research Center of Industrial Microorganisms, Fudan University, Shanghai, China, 3 State Key Laboratory of Oncology in South China, Collaborative Innovation Center for Cancer Medicine, Sun Yat-sen University Cancer Center, Guangzhou, China, 4 Shanghai Institute of Nutrition and Health, University of Chinese Academy of Sciences, Chinese Academy of Sciences, Shanghai, China, 5 Center for Molecular Medicine, Children's Hospital, Fudan University, Shanghai, China, 6 Shanghai Key Laboratory of Medical Epigenetics, International Laboratory of Medical Epigenetics and Metabolism, Ministry of Science and Technology, Institutes of Biomedical Sciences, Fudan University, Shanghai, China

\* fcctangjn@zzu.edu.cn (JT); ymw@fudan.edu.cn (YW)

## Abstract

CRISPR-Cas12a, often regarded as a precise genome editor, still requires improvements in specificity. In this study, we used a GFP-activation assay to screen 14 new Cas12a nucleases for mammalian genome editing, successfully identifying 9 active ones. Notably, these Cas12a nucleases prefer pyrimidine-rich PAMs. Among these nucleases, we extensively characterized Mb4Cas12a obtained from *Moraxella bovis CCUG 2133*, which recognizes a YYN PAM (Y = C or T). Our biochemical analysis demonstrates that Mb4Cas12a can cleave double-strand DNA across a wide temperature range. To improve specificity, we constructed a SWISS-MODEL of Mb4Cas12a based on the FnCas12a crystal structure and identified 8 amino acids potentially forming hydrogen bonds at the target DNA-crRNA interface. By replacing these amino acids with alanine to disrupt the hydrogen bond, we tested the influence of each mutation on Mb4Cas12a specificity. Interestingly, the F370A mutation improved specificity with minimal influence on activity. Further study showed that Mb4Cas12a-F370A is capable of discriminating single-nucleotide polymorphisms. These new Cas12a orthologs and high-fidelity variants hold substantial promise for therapeutic applications.

## Introduction

The RNA-guided CRISPR-Cas effectors serve as adaptive immune systems protecting bacteria and archaea from invasive foreign DNA [1] and have been employed for genome editing [2,3]. Notably, Type II Cas9 and Type V-A Cas12a are the most frequently used effectors for mammalian genome editing [4–10]. In contrast to CRISPR-Cas9, CRISPR-Cas12a stands out as

---

**Data Availability Statement:** All relevant data are within the paper and its Supporting Information files. The raw sequencing data for deep sequencing have been submitted to the NCBI Sequence Read Archive (PRJNA1055423). The raw data for the

---

GUIDE-seq Data have been submitted to the NCBI Sequence Read Archive (PRJNA1106073), and the raw data for Sanger sequencing were packaged in the compressed supplemental file S1 Data in the revised manuscript. The raw images underlying data about all blot and gel results can be found in S1 Raw Images.

**Funding:** This work was supported by grants from the National Key R&D Program of China (https://service.most.gov.cn/xmtj/) (2021YFC270110, 2021YFA0910602, Y.M); the National Natural Science Foundation of China (https://www.nsfc.gov.cn/english/site_1/index.html) (8270254, 82070258, Y.M); Open Research Fund of State Key Laboratory of Genetic Engineering, Fudan University (https://www.fudan.edu.cn/491/list.htm) (No. SKLGE-2104, Y.M), Science and Technology Research Program of Shanghai (https://svc.stcsm.sh.gov.cn/) (2ZR1426000, 19DZ2282100, Y.M). The funders had no role in study design, data collection and analysis, decision to publish, or preparation of the manuscript.

**Competing interests:** Wang Yongming have applied for the patent and the patent application ID was 202110606220 .9.

**Abbreviations:** ANOVA, analysis of variance; DMD, Duchenne muscular dystrophy; DMEM, Dulbecco's Modified Eagle Medium; DTT, dithiothreitol; GST, glutathione S-transferase; iPSC, induced pluripotent stem cell; NLS, nuclear localization signal; PFA, paraformaldehyde; PMSF, phenylmethanesulfonyl fluoride; PSP, PreScission protease; RFLP, restriction fragment-length polymorphism; SNV, single-nucleotide variant; TB, Terrific Broth; β-ME, β-mercaptoethanol.

particularly interesting due to its streamlined composition, consisting of only 2 essential components: a Cas12a nuclease and a CRISPR RNA (crRNA) [10]. Cas12a and crRNA form a complex that recognizes the target DNA that is complementary to the crRNA [11]. Cas12a processes the crRNA transcribed from the CRISPR array, a feature that dramatically simplifies multiplex genome editing [12]. In addition, Cas12a generates staggered ends upon DNA cleavage that may promote site-directed integration events [13]. In comparison to SpCas9, Cas12a exhibits higher specificity, often associated with fewer or even no detectable off-target effects [14,15]. These distinctive features collectively position Cas12a as an excellent choice for genome editing applications.

Cas12a has found widespread applications in both animals and plants. For instance, Zhang and colleagues utilized Cas12a to correct Duchenne muscular dystrophy (DMD) mutations in induced pluripotent stem cells (iPSCs) derived from patients and in mdx mice, an established model for DMD [16]. Dong and colleagues engineered Cas12a-expressing silkworms to counter nucleopolyhedrovirus infections [17]. Additionally, Zhang and colleagues developed a multiplexed Cas12a system, enabling the simultaneous editing of 6 sequences to enhance yield and blight disease resistance in rice [18]. Beyond these, Cas12a has been employed in creating rat and mouse models [19,20], as well as editing human hepatocytes [21].

Since its inception in 2015 [10], Cas12a has undergone significant development, leading to the identification and utilization of a dozen Cas12a orthologs for genome editing in human cells [22–26]. These orthologs exhibit diverse characteristics, encompassing editing activity, specificity, PAM requirements, and target sequence preferences [22,24]. This diversity broadens our capacity to efficiently edit a multitude of genomic sites. In the current study, we systematically screened 14 Cas12a orthologs and pinpointed Mb4Cas12a as a particularly efficient enzyme. Furthermore, we conducted engineering efforts on Mb4Cas12a to enhance its specificity. This research expands the CRISPR-Cas12a toolbox, offering a more versatile set of tools for genomic manipulation.

## Results

### A screen of active Cas12a orthologs by the GFP-activation assay

To identify new Cas12a orthologs for genome editing, we employed LbCas12a as a reference to search for related Cas12a orthologs in the NCBI database. We selected 14 Cas12a orthologs whose identity to LbCas12a ranged from 34.7% to 45.0% (Table 1). Further analysis of each function domain showed that the WED domain (playing the key role in crRNA recognition and processing) and the RuvC endonuclease domain had a higher identity to LbCas12a (S1A and S1B Fig). The protein lengths varied from 1,205 to 1,325 amino acids. Phylogenetic analysis showed that 4 orthologs clustered to the AsCas12a clade, 9 orthologs clustered to the FnCas12a clade, and 1 ortholog (Px2Cas12a) belonged to an independent clade (S2 Fig). We identified direct repeats for 12 Cas12a orthologs. These repeat sequences were highly conserved at the stem regions but less conserved at the loop regions (S3A Fig). Consequently, these mature repeats formed very similar secondary structures with the same stems but different loops (S3B Fig). These data indicated that these Cas12a orthologs might exchange the repeat sequences for genome editing.

Next, we tested Cas12a ortholog activity using a previously developed GFP-activation assay [5]. In this assay, a lentivirus vector carries a GFP reporter gene. The GFP coding sequence is disrupted by a DNA fragment containing 5 random sequences followed by a target sequence (protospacer) (Fig 1A). If a Cas12a ortholog can edit the target sequence and generate (insertions/deletions) indels, the in-frame mutation will occur in a portion of cells, resulting in GFP expression. The Cas12a orthologs underwent human codon optimization, synthesis, and the

**Table 1. Fourteen Cas12a orthologs were selected from the NCBI database.**

| Name | NCBI ID | Host strain | Length (aa) | Identity to LbCas12a |
|---|---|---|---|---|
| BgCas12a | OLA11341.1 | *Bacteroides galacturonicus DSM 3978* | 1,305 | 37.0% |
| Fb2Cas12a | WP_089081092.1 | *Flavobacterium branchiophilum DSM 24789* | 1,318 | 43.5% |
| Fn3Cas12a | WP_004339290.1 | *Francisella tularensis novicida FTG, UT01-4992* | 1,307 | 37.1% |
| FsCas12a | WP_045971446.1 | *Flavobacterium sp. 316* | 1,273 | 45.0% |
| HoCas12a | TFF64011.1 | *Helcococcus ovis* | 1,325 | 34.7% |
| Lp2Cas12a | WP_055306762.1 | *Lachnospira pectinoschiza 2789STDY5834886* | 1,305 | 37.3% |
| Mb4Cas12a | WP_078273923.1 | *Moraxella bovis CCUG 2133* | 1,261 | 41.5% |
| Mb5Cas12a | WP_046700744.1 | *Moraxella bovoculi 58069* | 1,251 | 39.9% |
| MeCas12a | WP_079324973.1 | *Moraxella equi CCUG 4950* | 1,251 | 40.6% |
| Ml2Cas12a | WP_065256572.1 | *Moraxella lacunata CCUG 57757A* | 1,264 | 40.8% |
| Ml3Cas12a | WP_115006085.1 | *Moraxella lacunata* | 1,261 | 37.5% |
| MoCas12a | WP_112744621.1 | *Moraxella ovis NCTC 11019* | 1,261 | 40.9% |
| PiCas12a | WP_161942329.1 | *Prevotella ihumii Marseille-P3385* | 1,320 | 43.6% |
| Px2Cas12a | WP_151622887.1 | *Pseudobutyrivibrio xylanivorans MA3014 v2* | 1,205 | 43.7% |

addition of nuclear localization signals (NLSs) at both ends. Subsequently, they were cloned into a mammalian expression plasmid. The Cas12a protein expression was confirmed by western blot (S4A Fig); the nuclear localization of each Cas12a ortholog was confirmed by immunofluorescence staining (S4B Fig). We transfected Cas12a and corresponding crRNA expression plasmids into cells (Fig 1B). FsCas12a and MeCas12a used the LbCas12a crRNA scaffold for genome editing. LbCas12a was used as a positive control. Five days after transfection, GFP-positive cells were observed for LbCas12a and 9 new Cas12a orthologs (Fig 1C). These new orthologs hold potential for genome editing.

## Analysis of PAMs

To analyze PAMs, we sorted out GFP-positive cells and extracted genomic DNA. The 5-bp random sequences were PCR-amplified for deep sequencing. The sequencing results showed that in-frame mutations occurred (Fig 2A). We generated PAM logos and wheels based on deep sequencing data (Fig 2B and 2C). LbCas12a strongly preferred a TTTV PAM (V = A, C, or G), consistent with a previous study [10]. All new Cas12a orthologs recognized degenerate PAMs. They strongly preferred T, followed by C at PAM positions −2 and −3. They slightly preferred C and G at the PAM position −1. BgCas12a, HoCas12a, and PiCas12a also displayed sequence biases at the PAM position −4.

Next, we tested the activity of BgCas12a, Mb4Cas12a, Ml2Cas12a, and MoCas12a because they induced more GFP-positive cells. We designed a panel of 16 endogenous targets with TTN, CCN, CTN, and TCN PAMs. Five days after transfection of Cas12a and crRNA expressing plasmids, genomic DNA was extracted for targeted deep sequencing. The sequencing results showed that indels occurred at these sites with varied efficiency depending on the loci (S5A Fig). Interestingly, these orthologs displayed different target sequence preferences at some loci, such as TS1, TS2, and TS3, offering opportunities for site-specific genome editing. Overall, these 4 orthologs displayed comparable activity (S5B Fig). These data demonstrated that these 4 orthologs enabled genome editing with the YYN PAM (Y = C, or T).

We further characterized Mb4Cas12a PAMs by the in vitro cleavage assay. We designed a target with 8 different PAMs, including TTA, TTC, TCA, TCC, CTA, CTC, CCA, and CCC PAMs. The Cas12a nucleases purified from *Escherichia coli* cells were incubated with in vitro-transcribed crRNAs and target DNA substrates. After 8 hours of incubation at 37˚C, the

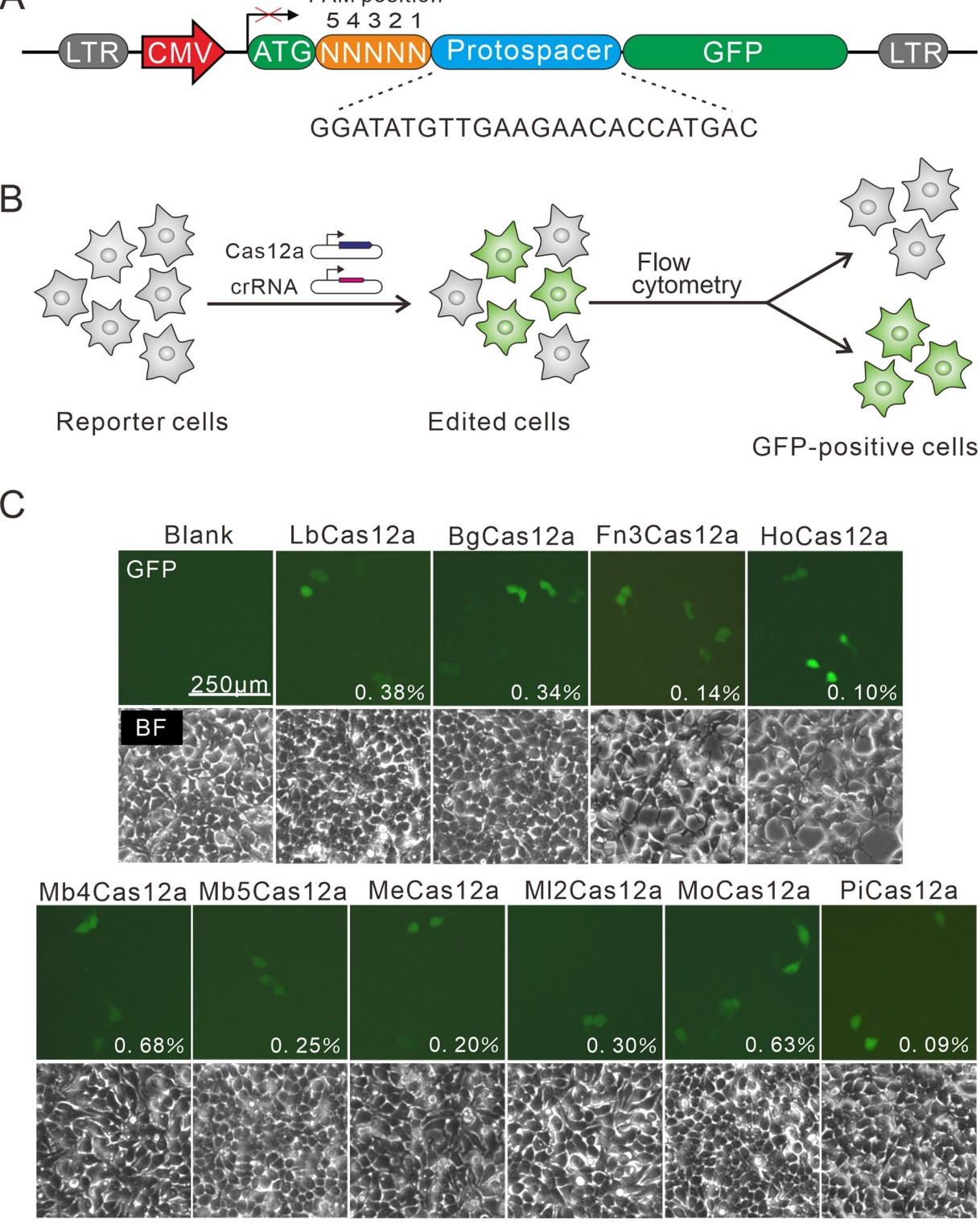

**Fig 1. Analysis of Cas12a ortholog activities. (A)** Schematic of the GFP-activation assay. A 5-bp random sequence followed by a 24-bp protospacer (target sequence) is inserted between ATG and the GFP-coding sequence. The library is carried by a lentivirus vector. **(B)** crRNA/Cas12a-expressing plasmids are transfected into reporter cells. The GFP-positive cells are sorted out, and target sequences are PCR-amplified for deep sequencing. **(C)** Cas12a orthologs induce GFP expression. The percentage of GFP-positive cells was shown. The cells without transfection of Cas12a were used as a negative control. BF, bright field; GFP, green fluorescent protein.

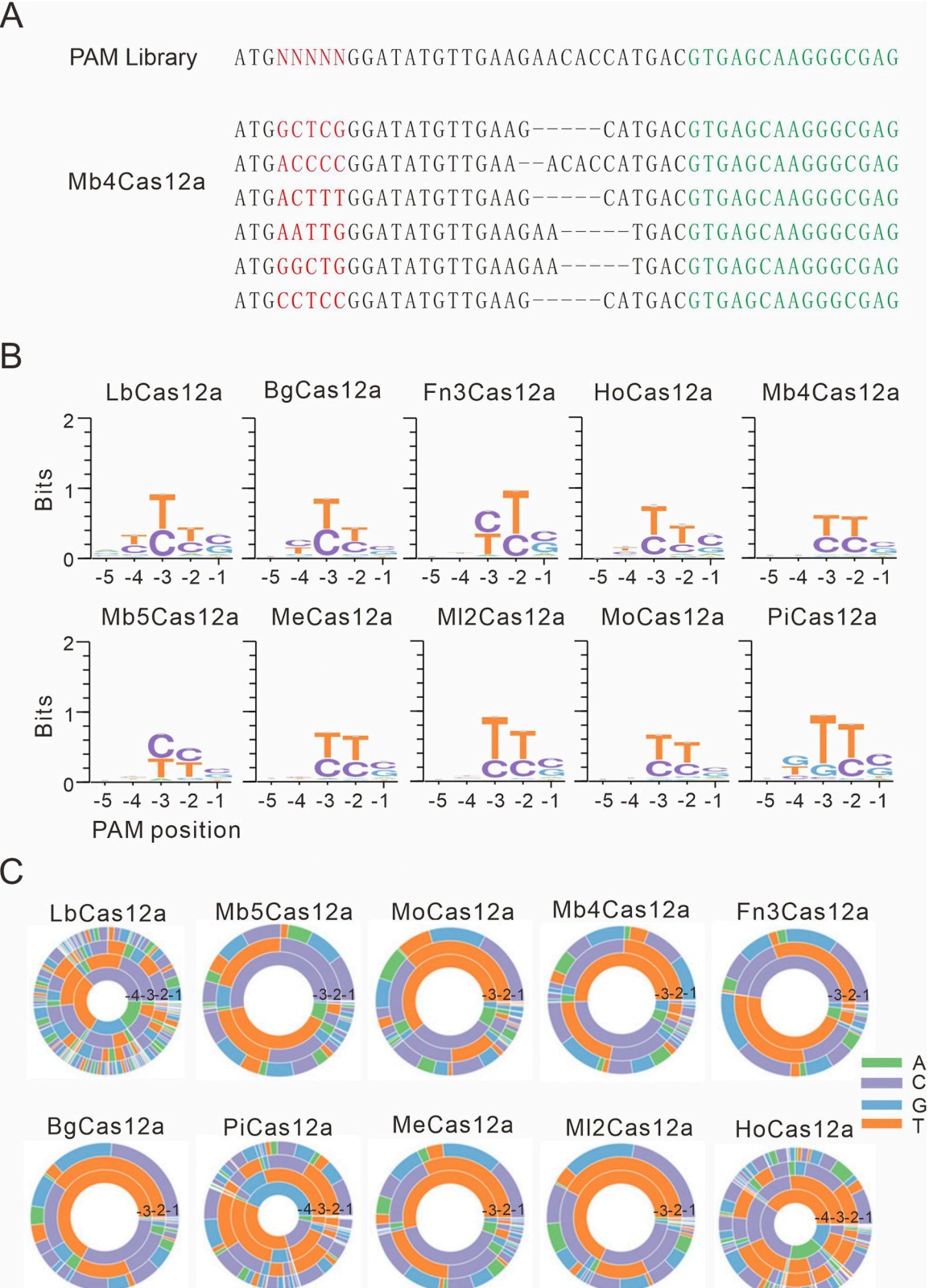

**Fig 2. Analysis of Cas12a PAMs.** (**A**) Deep sequencing reveals that Mb4Cas12a induced indels on the target site. PAM is shown in red; the GFP sequence is shown in green. (**B**) PAM WebLogos of Cas12a is generated based on deep sequencing data. (**C**) PAM wheels of Cas12a are generated based on deep sequencing data. PAM positions are shown on the wheel pictures.

electrophoresis analysis showed that Mb4Cas12a could efficiently cleave targets with TTN, TCN, and CTN PAMs but less efficiently cleave targets with the CTA and CCA PAMs (Fig 3A). Next, we tested the optimal temperature of Mb4Cas12a cleavage. The Cas12a nucleases were incubated with in vitro-transcribed crRNAs and target DNA substrates for 1 hour. The electrophoresis analysis showed that Mb4Cas12a functioned over a wide range of temperatures with optimal cleavage efficiency at temperatures of 10˚C to 50˚C (Fig 3B). We analyzed the cleavage site of Mb4Cas12a using Sanger sequencing of the cleaved DNA ends. The results showed that Mb4Cas12a generated a 5-nt 5' overhang (Fig 3C), consistent with a previous study [10]. In the following study, we focused on Mb4Cas12a.

## Efficient genome editing with Mb4Cas12a

Next, we compared Mb4Cas12a activity to LbCas12a. They were expressed with the same construct backbone (Fig 4A), and similar protein expression levels were observed (Fig 4B). We designed a panel of 13 endogenous loci with TTTV PAMs. Five days after transfection of Cas12a and crRNA expressing plasmids, genomic DNA was extracted for targeted deep sequencing. The sequencing results showed that LbCas12a displayed similar or higher activity than Mb4Cas12a at most of the loci, except the locus E5, where Mb4Cas12a displayed higher activity than LbCas12a (Fig 4C and 4D).

We tested Mb4Cas12a activity in additional cell lines, including human HeLa, HCT116, A375, and SH-SY5Y cells. We designed a panel of 18 endogenous loci with TTN, CCN, CTN, and TCN PAMs. Mb4Cas12a enabled genome editing at all loci with varied efficiency (S6A–S6D Fig). We further tested Mb4Cas12a activity in mouse N2a cells with a panel of 9 endogenous loci with the TTN PAM. Mb4Cas12a enabled genome editing at all loci (S6E Fig). These data demonstrated that Mb4Cas12a enabled genome editing in a variety of cell types.

## Analysis of Mb4Cas12a specificity

Next, we used the GFP-activation assay to analyze Mb4Cas12a specificity, and LbCas12a was used as a control. We designed a panel of crRNAs with dinucleotide mutations. If the off-target cleavage occurs, the GFP-positive cells will be observed. The results showed that both Mb4Cas12a and LbCas12a were sensitive to the dinucleotide mismatches at crRNA positions 1 to 20 (S7A Fig). Next, we performed an in vitro cleavage assay to analyze Mb4Cas12a specificity. After 8 hours of incubation of Mb4Cas12a with in vitro-transcribed crRNAs and target DNA substrates at 37˚C, the electrophoresis analysis showed that Mb4Cas12a was sensitive to the dinucleotide mismatches (S7B Fig). Next, we tested Mb4Cas12a specificity by designing a panel of crRNAs with single-nucleotide mismatches. The GFP-activation assay showed that LbCas12a substantially tolerated single-nucleotide mismatches, while Mb4Cas12a less tolerated single-nucleotide mismatches (Fig 5A and 5B).

A previous study has hypothesized that the SpCas9-sgRNA complex may possess more energy than is required for optimal recognition of its intended target DNA site, potentially facilitating the cleavage of mismatched off-target sites [27]. This phenomenon is partially attributed to certain Cas9 amino acid residues forming hydrogen bonds with the target DNA. Disrupting these hydrogen bonds has been shown to increase the specificity of Cas9 [16,27,28]. To improve Mb4Cas12a specificity, we analyzed the FnCas12a crystal structure and identified 8 amino acid residues that potentially form hydrogen bonds at the target DNA-crRNA interface (S8A Fig). We constructed a SWISS-MODEL of Mb4Cas12a based on the FnCas12a crystal structure and identified corresponding residues (S8A Fig). Seven amino acid residues can also be identified by sequence alignment of Mb4Cas12a to FnCas12a (S8B Fig). SWISS-MODEL of Mb4Cas12a showed that F370 instead of Y381 corresponded to FnCas12a Y410 (S8C Fig).

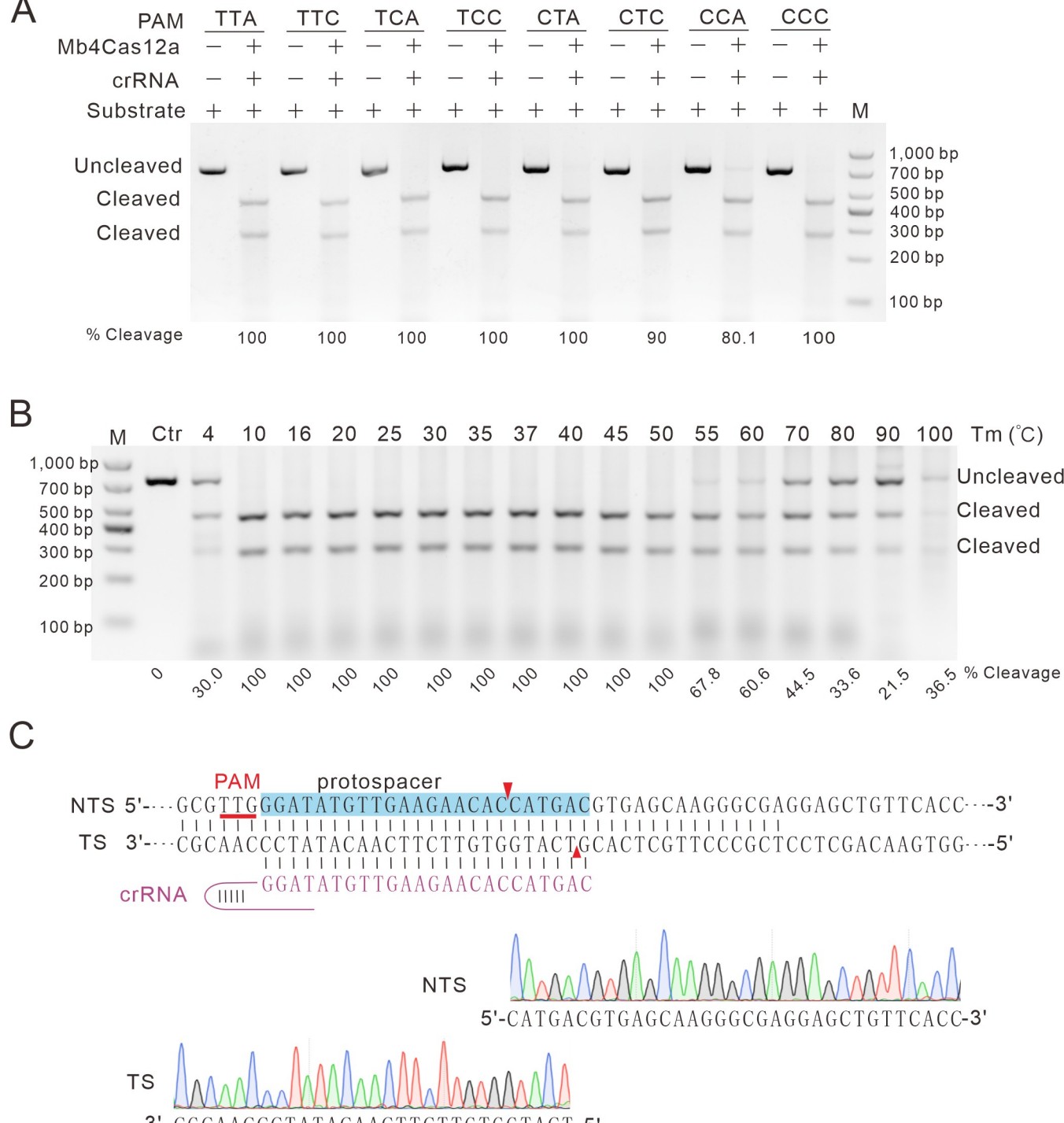

**Fig 3. Characterization of Mb4Cas12a with the in vitro cleavage assay.** (**A**) DNA cleavage of Mb4Cas12a at 37°C for 8 hours. The PAMs are shown on the top; the cleaved bands are indicated by green asterisks; the cleavage efficiency is shown below. (**B**) DNA cleavage of Mb4Cas12a at different temperatures for 1 hour. Temperatures are shown on the top; the cleavage efficiency is shown below. Ctr, control, the DNA fragment without incubation with Mb4Cas12a and crRNA. (**C**) Sanger sequencing traces from Mb4Cas12a-cleaved target show staggered 5-nt 5' overhang. NTS, nontarget site; TS, target site. The cleaved target sites are indicated by red triangles.

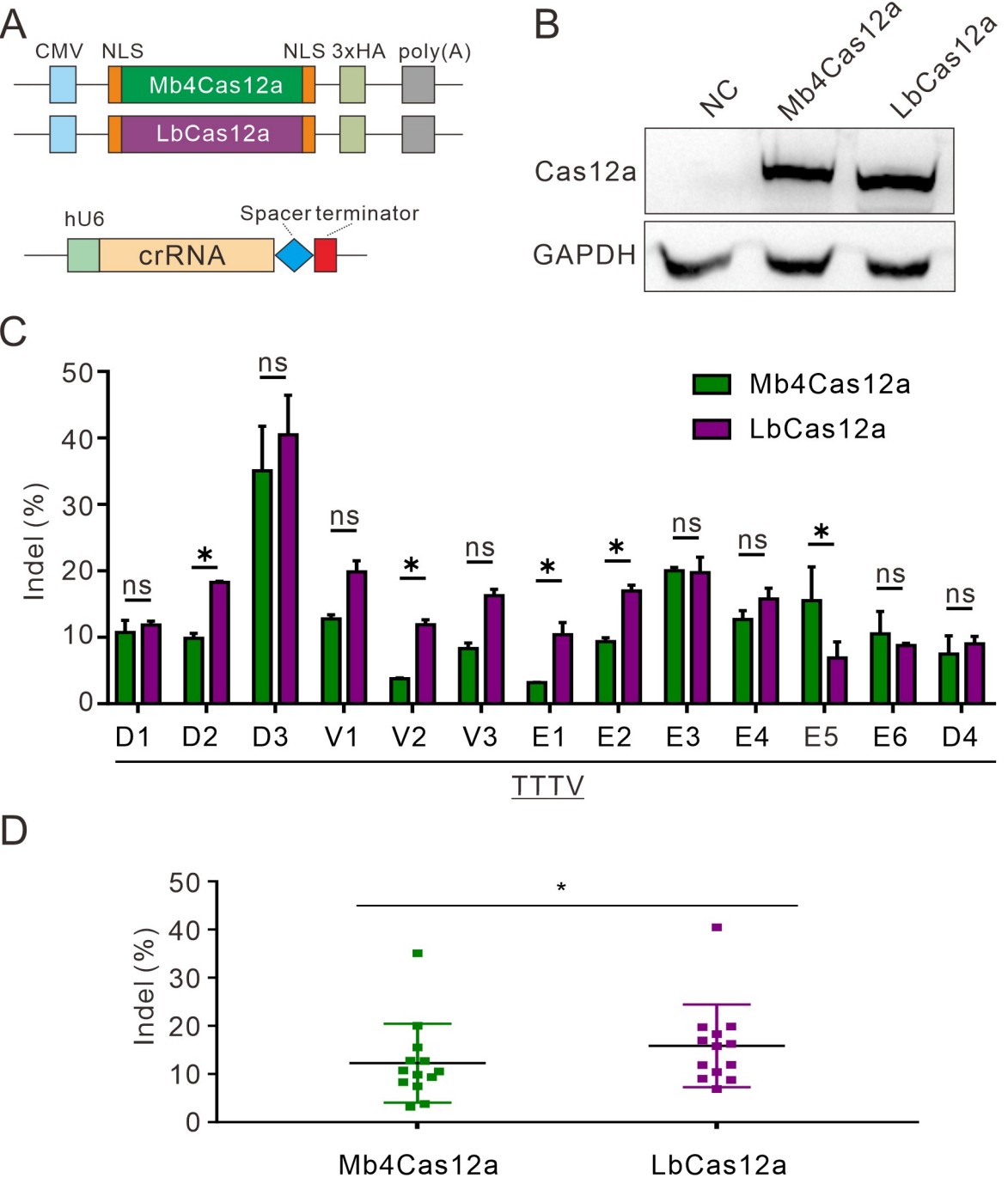

**Fig 4. Genome editing with Mb4Cas12a at endogenous sites.** (**A**) Schematic of the Cas12a and crRNA expression constructs. (**B**) The protein expression level of Cas12a was measured by western blot. Cells without Cas12a transfection were used as a negative control. (**C**) Comparison of Mb4Cas12a and LbCas12a activity at 13 endogenous loci ($n = 3$). The data represent the mean ± SD. Student $t$ test, ns, means no significance, $^*$ $p < 0.05$. (**D**) Quantification of editing efficiency for Mb4Cas12a and LbCas12a. The data represent the mean ± SD; $n = 3$. Student $t$ test, $^*$ $p < 0.05$. The numerical values underlying this figure can be found in S5 Table.

The individual residue was replaced by alanine and tested the variant specificity with the GFP-activation assay. The results revealed that 2 mutations (G305A and S307A) could mildly improve Mb4Cas12a specificity, and 4 mutations (F370A, G257A, N276A, and K291A) could improve Mb4Cas12a specificity (Figs 5A, 5B and S9A–S9H). We tested the influence of the latter 4 mutations on Mb4Cas12a activity at 2 endogenous loci. The results showed that these mutations decreased editing activity (Fig 5C and 5D). In the following study, we focused on Mb4Cas12a-F370A due to its high specificity.

Next, we performed GUIDE-seq to test the genome-wide off-target effects of Mb4Cas12a and F370A. LbCas12a was used as a control. We selected 10 targets that LbCas12a or AsCas12a can robustly generate off-targets from literature [25,29]. The Cas12a/crRNA expressing plasmids and the GUIDE-seq oligos were transfected into HEK293T cells. Five days after transfection, we prepared libraries for deep sequencing. The sequencing results revealed that on-target cleavage occurred for all 3 Cas12a nucleases at all loci, as revealed by the high GUIDE-seq read counts (Figs 6A, 6B and S10). LbCas12a generated 59 off-targets across 10 tested loci. In contrast, Mb4Cas12a generated 24 off-targets across 10 tested loci, while F370A generated 19 off-targets across 10 tested loci. These data demonstrate that both Mb4Cas12a and F370A are more specific than the canonical LbCas12a.

The majority of off-targets were produced at the DNMT1-sg3 and POLQ sites. When targeting the DNMT1-sg3 site, Mb4Cas12a induced 18 off-target sites, while F370A produced 12 off-target sites (Fig 6A). Mb4Cas12a demonstrated higher specificity than F370A at 7 off-target sites, whereas F370A exhibited greater specificity than Mb4Cas12a at 18 off-target sites, based on reads number. Interestingly, we observed a bias in the off-targeting behaviour of Mb4Cas12a and its F370A variant when targeting the POLQ locus. For example, Mb4Cas12a exhibited higher specificity than F370A at the off1 site, whereas F370A displayed greater specificity than Mb4Cas12a at the off4 site (S10A Fig). Similarly, at the off2 and off7 sites, which differed by only 1 nucleotide in the PAM region, Mb4Cas12a showed greater specificity than F370A at the off2 site, while F370A demonstrated higher specificity than Mb4Cas12a at the off7 site. These observations suggest potential variations in the off-targeting specificity of different Mb4Cas12a variants. In future studies, it would be intriguing to investigate further the specificity of other Mb4Cas12a variants using GUIDE-seq.

## Allele-specific genome editing with Mb4Cas12a-F370A

Finally, we investigated the capability of Mb4Cas12a-F370A for allele-specific genome editing. We previously discovered 4 SNPs in the HEK293T cells [30]. We designed crRNAs targeting these loci with SNPs at the mismatch-sensitive region. The Mb4Cas12a-F370A and crRNA-expressing plasmids were transfected into HEK293T cells. Five days after transfection, genomic DNA was extracted for targeted deep sequencing. The sequencing results revealed that Mb4Cas12a-F370A could efficiently generate indels at the target allele with minimal off-target effects at the nontarget allele (Fig 7A–7D). For the rs45545732 locus, strikingly, Mb4Cas12a-F370A resulted in an indel rate of 26.4% at the target allele with a minimal off-target rate of 1.2% at the nontarget allele. To test whether Mb4Cas12a-F370A enabled allele-specific genome editing in other cell lines, we repeated the experiments in a neuroblastoma cell line (SH-SY5Y) and a human cervical cancer cell line (C33A). The results revealed that Mb4Cas12a-F370A enabled allele-specific genome editing in these 2 cell lines (S11 and S12 Figs).

In silico analysis revealed that Mb4Cas12a-F370A could allele-specifically disrupt 35,906 clinically relevant variants in the ClinVar database [31] and 593,719,868 SNPs in the dbSNP database by using a previously established computational pipeline (Fig 7E) [32]. The crRNAs

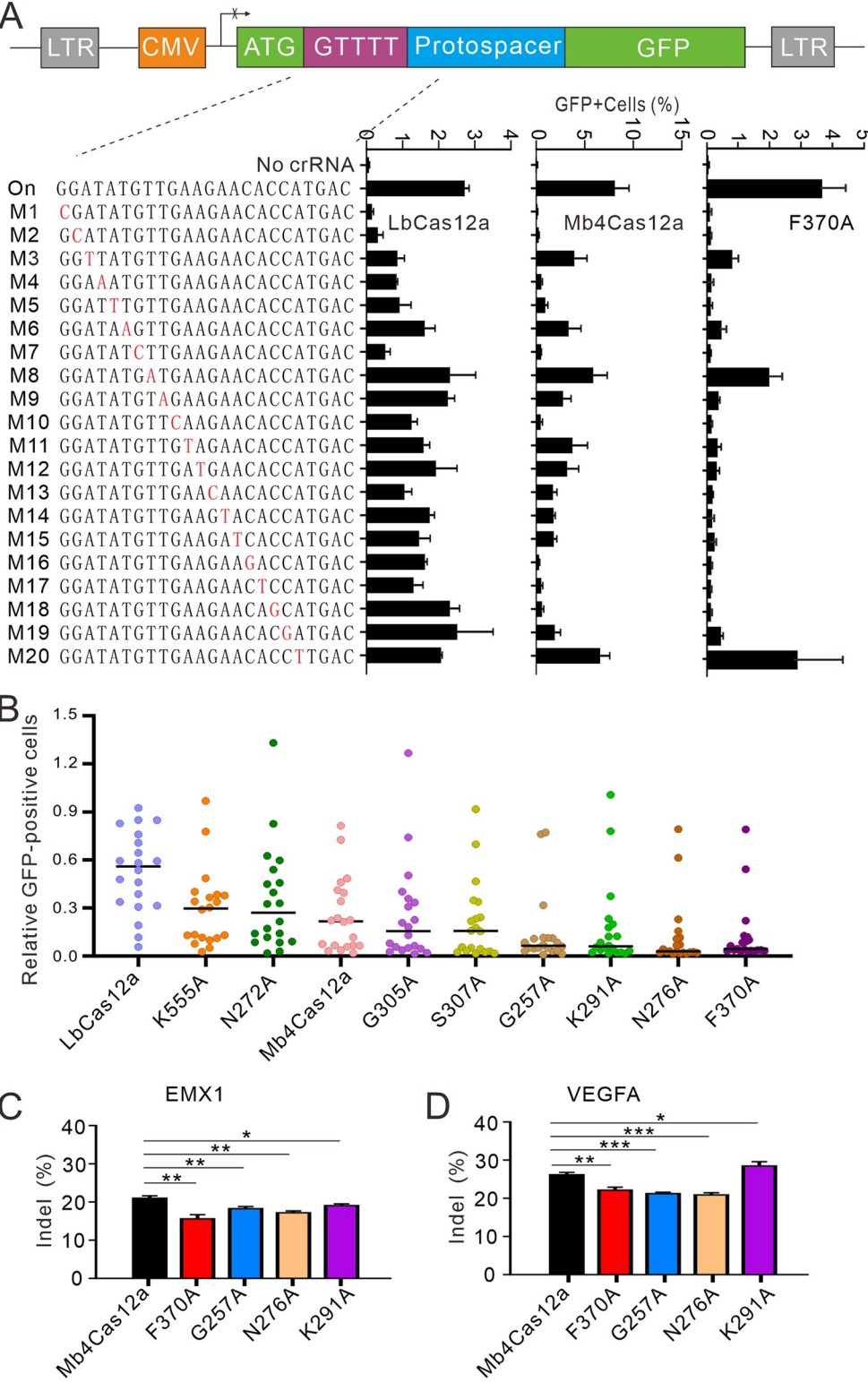

**Fig 5. Test of Mb4Cas12a specificity by the GFP activation assay.** (**A**) The GFP-activation assay shows that Mb4Cas12a and LbCas12a substantially tolerate single-nucleotide mismatches. A schematic of the GFP-activation reporter is shown on the top; crRNAs are shown on the left; single-nucleotide mismatches are indicated in red. On, on-target crRNA. (**B**) Quantification of off-target efficiency. The off-target efficiency is normalized by on-target efficiency. (**C**, **D**) Test of 4 Mb4Cas12a variant activity at 2 endogenous loci *EMX1* (**C**) and *VEGFA* (**D**). The data represent the

mean ± SD; $n$ = 3. Student $t$ test, * $p < 0.05$. ** $p < 0.01$. *** $p < 0.001$. The numerical values underlying this figure can be found in S5 Table.

designed for clinically relevant variants were deposited in S4 Table. Users can select these crRNAs from the database to disrupt clinically relevant mutations.

## Discussion

Cas12a is the second most frequently used CRISPR-Cas effector for mammalian genome editing [10]. Although several Cas12a orthologs have been developed for mammalian genome editing [22], it is still useful to develop new Cas12a tools because these orthologs prefer different target sequences [25,33]. In this study, we showed that BgCas12a, Mb4Cas12a, Ml2Cas12a, and MoCas12a enabled genome editing in mammalian cells, expanding the Cas12a toolbox. We also showed that these orthologs preferred different target sequences, allowing site-specific genome editing. Considering the short degenerate PAM requirement, these orthologs allow genome editing of a large scope of genomic loci.

Importantly, we developed a high-fidelity version of Mb4Cas12a variants. Cas12a displays higher specificity than SpCas9, with fewer or no detectable off-target effects depending on the target [14,15]. LbCas12a is one of the most frequently used Cas12a, but it tolerates single-nucleotide mismatches. So far, however, no Cas12a orthologs have been shown to discriminate SNPs. In this study, we demonstrate that Mb4Cas12a is more specific than LbCas12a. We further demonstrate that the high-fidelity version of Mb4Cas12a-F370A can discriminate SNPs and enable allele-specific disruption of target alleles. Several studies have shown the potential to treat autosomal dominant diseases by allele-specific disruption of mutant alleles through NHEJ [34,35]. Mb4Cas12a-F370A offers a new genome editing tool for the treatment of such diseases.

## Materials and methods

### Plasmid construction

**Plasmids for Cas12a orthologs or crRNA expression.** The plasmid pAAV-CMV-notracr vector was amplified by primers pAAV-notracr-F/pAAV-notracr-R using pAAV-CMV-Sauri-Cas9-puro (Addgene No. 135965) as a template. The human codon-optimized Cas12a gene was synthesized by HuaGene (Shanghai, China) and cloned into the pAAV-CMV-notracr backbone by the NEBuilder assembly tool (NEB). The sequence of each Cas12a protein was confirmed by Sanger sequencing (GENEWIZ, Suzhou, China). Oligonucleotide duplexes corresponding to Cas12a orthologs and spacer sequences were PCR amplified and ligated into pSK-mU6-Cj-sgRNA-SV40-puro plasmids (Addgene No. #192128) for mU6 promoter-driven expression of crRNAs. The sequences of the Cas12a gene and crRNA used in this study are listed in S1 Table.

**Cell culture and transfection.** All cells were grown in humidified 37°C, 5% $CO_2$ incubators. HEK293T, HeLa, A375, SH-SY5Y, and N2a cells were cultured in Dulbecco's Modified Eagle Medium (DMEM). HCT116 cells were maintained in McCoy's 5A. All the cells were supplemented with 10% FBS (Gibco), 100 U/mL penicillin, and 100 mg/mL streptomycin. All cell lines used were tested negative for *Mycoplasma* contamination.

A day before transfection, cells were trypsinized and seeded to each well of the 48-well plate. For transient transfection, cells were transfected with Cas12a plasmids (300 ng) + sgRNA plasmids (200 ng) by Lipofectamine 2000 or 3000 (1 μL) (Life Technologies) according to the manufacturer's instructions. HEK293T, HeLa, and N2a cells were transfected with

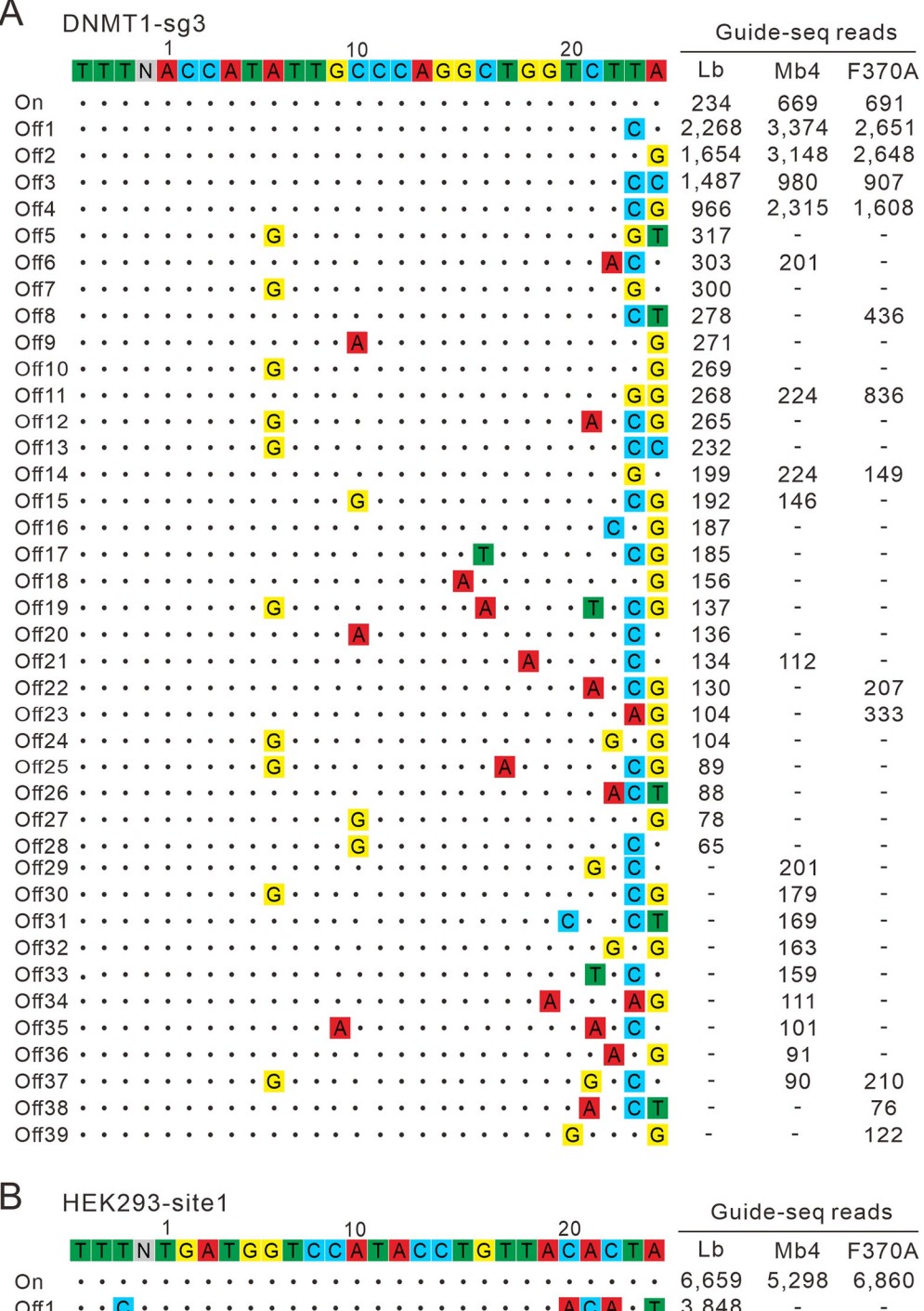

**Fig 6. Test of Cas12a specificity with GUIDE-seq. (A, B)** The genome-wide off-target effects of LbCas12a, Mb4Cas12a, and F370A are analyzed by GUIDE-seq. On-target and off-target sequences are shown on the left. Read numbers for on-target and off-target sites are shown on the right. The reads number for off-target sites less than 50 generated by LbCas12a is not displayed. Mismatches compared to the on-target site are shown and highlighted in colour.

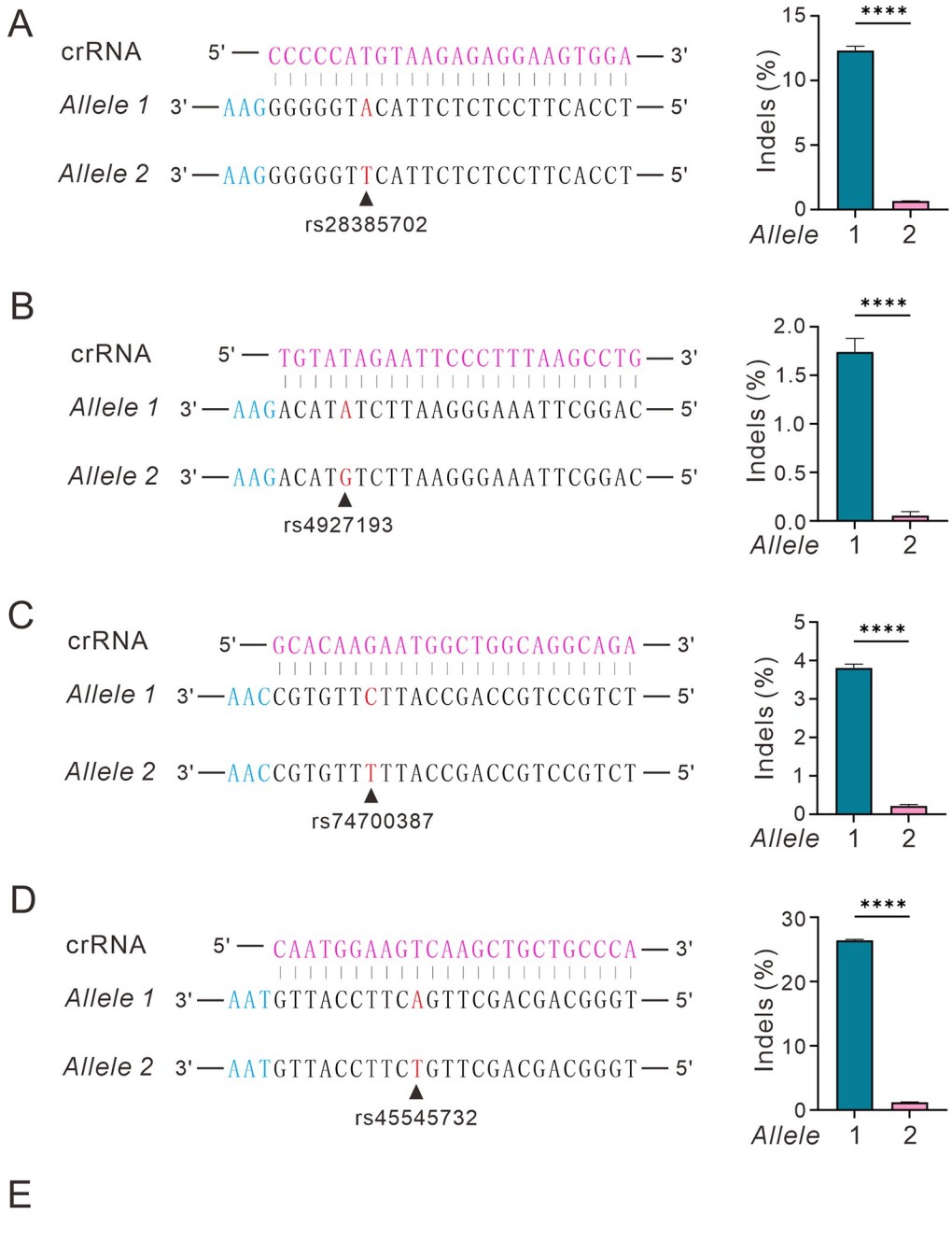

**Fig 7. Mb4Cas12a-F370A allows allele-specific genome editing.** (**A**, **B**) Mb4Cas12a-F370A allele specifically disrupts the single-nucleotide polymorphism (SNP) at locus rs28385702 (**A**), rs4927193 (**B**), rs747000387 (**C**), rs45545732 (**D**). PAMs are shown in light blue; target SNPs are shown in red; crRNAs are shown in purple; indel efficiencies are shown on the right. Data represent mean ± SD. $n = 3$. (**E**) Summary of pathogenic sites or SNPs that are potentially disrupted by Mb4Cas12a-F370A. The data represent the mean ± SD; $n = 3$. Student $t$ test, **** $p <0.0001$. The numerical values underlying this figure can be found in S5 Table.

Lipofectamine 2000 (Life Technologies). SH-SY5Y, A375, and HCT116 cells were transfected with Lipofectamine 3000 (Life Technologies). Cells were collected 5 days after transfection. Genomic DNA was isolated, and the target sites were PCR-amplified by nested PCR amplification and purified by a Gel Extraction Kit (QIAGEN) for deep sequencing. The primer sequences for deep sequencing are listed in S2 Table. The guide RNA sequences used in this article are listed in S3 Table.

**Phylogenetic analysis.** Phylogenetic analysis was conducted using the vector NTI software, employing its default parameters. The Clustal W algorithm served as the alignment method. In essence, an initial assessment of crude similarity among all sequence pairs, termed "Parities alignment," is conducted. These similarity scores are subsequently employed in generating a "guide tree" or dendrogram. This dendrogram dictates the order in which sequences are aligned during the final multiple-alignment stage. Following the calculation of the dendrogram, sequences are aligned progressively, starting with smaller groups and expanding until the entirety of sequences is encompassed in the conclusive alignment. This method ensures a systematic and comprehensive approach to multiple sequence alignment.

**Construction of GFP activation system for PAM detection.** The reporter plasmid was constructed as previously described with minimal modifications [36]. Briefly, The DNA oligonucleotide (NNNNNGGATATGTTGAAGAACACCATGAC) with 20-nt flanking homologous sequences (AAGCCTTGTTTGCCACCATG/GTGAGCAAGGGCGAGGAGCT) for Gibson Assembly were synthesized by GENEWIZ (Suzhou, China). Subsequently, full-length oligonucleotides were PCR-amplified utilizing Q5 High-Fidelity 2X Master Mix (NEB), followed by size selection on a 3% agarose gel EX (Life Technologies, Qiagen) and purification using the MinElute Gel Extraction Kit (Qiagen). The PCR products were then cloned into a lentiviral vector through Gibson Assembly (NEB) and further purified with Agencourt AMPure XP SPRI beads (Beckman Coulter). Electroporation of the Gibson Assembly products into MegaX DH10BTM T1R ElectrocompTM Cells (Invitrogen) was carried out using a Gene-Pulser (BioRad). Subsequently, the bacteria were introduced into recovery media and allowed to grow at 32˚C with agitation at 250 rpm overnight. Plasmid DNA was then purified from the bacteria using the Endotoxin-Free Plasmid Maxiprep kit (Qiagen).

**PAM sequence analysis.** The reporter plasmids were packaged into lentivirus and transfected into HEK293T cells as previously described [36]. Five days after transfection of Cas12a (10 μg) and gRNA-expressing plasmids (5 μg) into cells, genomic DNA was isolated for targeted deep sequencing of the random region. For PAM analysis, the 20-nt oligonucleotide flanking the target sequence was used to fix the target sequence. The sequences TTGTTTGCCACCATG and GGA were employed to fix the 5-nt random sequence. Target sequences with in-frame mutations were utilized for PAM analysis. The 5-nt random sequence was extracted and visualized using WebLogo3 [37] and a PAM wheel chart [38] to illustrate PAMs.

**Western blot analysis.** HEK293T cells were seeded into 24-well plates and transfected with Cas12a-expressing plasmid (800 ng) using Lipofectamine 2000 (Invitrogen). Three days after transfection, cells were collected and resuspended in lysis buffer for western blotting (Beyotime) supplemented with 1 mM phenylmethanesulfonyl fluoride (PMSF) (Beyotime). Cell lysates were then centrifuged at 12,000 rpm for 20 minutes at 4˚C, and the supernatants were collected for western blotting. An equal amount of protein samples was separated by SDS-PAGE gel and transferred to PVDF membranes. After blocking, membranes were probed with the anti-HA antibody (1:1,000; Abcam) and anti-GAPDH antibody (1:1,000; Cell Signaling) at 4˚C overnight. The membranes were washed 3 times in TBS-T for 5 minutes each time and incubated in the secondary goat anti-rabbit antibody (1:10,000; Abcam) for 1 hour at room temperature. The membranes were then washed with TBST buffer 3 times and imaged.

**Immunofluorescence staining.** For the efficient nuclear localization of these Cas12a variants, HEK293T cells were seeded into 24-well plates and transfected with Cas12a-expressing plasmid (800 ng) using Lipofectamine 2000 (Invitrogen). Three days after transfection, some cells were seeded into a 35-mm confocal dish. Next day, cells were fixed for 30 minutes in 4% paraformaldehyde (PFA) in PBS and permeabilized with 0.3% triton X-100 for 20 minutes at room temperature, followed by blocking with 5% BSA in PBS for 1 hour and then incubated with anti-HA antibody (Abcam, ab18181) overnight at 4˚C. After 3 washes, cells were incubated with goat anti-mouse FITC 1:300 (Abcam, ab6785) for 1 hour at room temperature. After 3 washes, the cells were imaged with a Zeiss LSM 880 confocal microscope.

**Protein expression and purification.** The *E. coli* expression plasmid of Mb4Cas12a containing a 6xHis-tag at N-terminal followed by a cleavage site for PreScission protease (PSP) were transformed in *E. coli Rosseta* (DE3) cells. Transformed bacteria were cultured into Terrific Broth (TB) medium at 37˚C before being induced with 0.1 mM isopropyl-1-thio-β-d-galactopyranoside at an OD600 of 0.6 and grown at 16˚C for 16 to 18 hours. Bacteria were lysed in 50 mM HEPES (pH 7.5), 300 mM NaCl, 30 mM imidazole, 1 μM DNase I, 1 mM PMSF, and 2 mM β-mercaptoethanol (β-ME) using a cell disruptor (JNBIO) and subjected to centrifugation at 20,000 × rpm for 90 minutes. The supernatant was filtered and applied to the first Ni-NTA column (GE Healthcare) equilibrated with binding buffer (20 mM HEPES (pH 7.5), 300 mM NaCl, 30 mM imidazole, and 2 mM β-ME). After being washed with wash buffer (20 mM HEPES (pH 7.5), 300 mM NaCl, 30 mM imidazole, and 2 mM β-ME), proteins were eluted with elution buffer (20 mM HEPES (pH 7.5), 300 mM NaCl, 300 mM imidazole, and 2 mM β-ME). The eluted protein was incubated with glutathione S-transferase (GST)-fused PSP to remove the 6xHis-tag and dialyzed overnight against dialysis buffer (20 mM HEPES (pH 7.5), 300 mM NaCl, and 2 mM β-ME). After dialysis, PSP was removed using a GST column. The protein was reapplied to the second Ni-NTA column equilibrated with dialysis buffer. Wash buffer was used to elute the proteins, which were subsequently loaded onto a Superdex 200 pg column (GE Healthcare) equilibrated with gel filtration buffer (20 mM HEPES (pH 7.5), 300 mM NaCl, and 1 mM dithiothreitol (DTT)). The proteins were eluted in a discrete peak corresponding to a molecular mass of approximately 170 kDa and stored at −20˚C. Then, the protein can be either used directly for biochemical assays or frozen at −80˚C. The sequence of bacterial expression plasmid is listed in S2 Table.

**In vitro transcription of crRNAs.** All crRNAs used in vitro cleavage experiments were synthesized using HiSribe T7 Quick High Yield RNA Synthesis Kit (NEB) according to the manufacturer's instructions. ssDNA oligos corresponding to the reverse complement of the target RNA sequence and a short T7 priming sequence were synthesized and annealed to obtain the template of the transcription reaction. T7 transcription was performed for 16 hours, and then RNA was purified using by RNA Clean XP (Beckman) beads and stored at −80˚C. The ssDNA oligos used for crRNA transcription are listed in S2 Table.

**In vitro cleavage assay.** For in vitro transcription, dsDNA substrates consisting of TTN nucleotides upstream of the protospacer were generated by PCR amplification. Cleavage in vitro was performed with purified proteins and corresponding crRNAs at 37˚C in cleavage buffer. Approximately 200 ng DNA substrate was incubated with 250 nM Mb4Cas12a protein, 500 ng crRNA, and 0.3 μL RNase inhibitor in 1x NEB Buffer 3.1 (100 mM NaCl, 50 mM Tris-HCl, 10 mM $MgCl_2$, and 100 μg/mL BSA). The cleavage products were denatured by incubating with 5 μg RNase A to digest crRNAs at 37˚C for 20 minutes and 1 μL proteinase K to degrade Cas12a protein at room temperature for 10 minutes. The cleavage products were analyzed by 2% agarose gel electrophoresis. For the functional time of Mb4Cas12a, the cleavage reactions were performed at 37˚C for 8 hours in a cleavage buffer. To determine the thermostability of Mb4Cas12a, the cleavage was reacted at a large-range temperature (4˚C to 100˚C)

for 1 hour in the cleavage buffer. Where applicable, quantitation of DNA cleavage or nicking was determined by the formula cleavage (%) = $100 \times (1\text{-sqrt}(1 - (b + c) / (a + b + c)))$, where a is the integrated intensity of the undigested products, and b and c are the integrated intensities of each cleavage or nicking products.

**Generation of the Mb4Cas12a variants.** The human codon-optimized Mb4Cas12a variants were cloned by circular PCR reaction. Briefly, wild-type Mb4Cas12a was used as the template to amplify PCR fragments using primers containing the mutation site. The primers are listed in S2 Table.

**Test of Cas12a specificity.** To test the specificity of LbCas12a, Mb4Cas12a, and its variants, we generated a GFP-reporter cell line with 5′-NTTTN (GTTTT) PAM. HEK293T cells were seeded into 48-well plates and transfected with Cas12a plasmids (300 ng) and crRNA plasmids (200 ng) by using Lipofectamine 2000 (1 μL). Five days after transfection, the GFP-positive cells were analyzed using the Calibur instrument (BD). Data were analyzed using the FlowJo software.

**GUIDE-seq.** GUIDE-seq experiments were performed as previously described [39]. Briefly, on the day of the experiment, 100 pmol of the double-stranded oligonucleotide (dsODN) GUIDE-seq tag was transfected into HEK293T cells with 500 ng Cas12a plasmid, 500 ng crRNA plasmid through the Neon Transfection System. The electroporation voltage, width, and number of pulses were 1,150 V, 20 ms, and 2 pulses, respectively. Genomic DNA was extracted with the DNeasy Blood and Tissue kit (QIAGEN) 5 to 7 days after electroporation. Approximately 500 ng of genomic DNA from each sample were randomly fragmented by NEBNext Ultra II FS DNA Module (NEB). A cleanup with 35 μL of AMPure XP beads (1X ratio) is performed according to manufacturer protocol and eluted in 12 μL of ddH2O. Around 10 μL end-repaired DNA was incubated with 10 μL NEBNext Adapter (E7335L), and 5 μL Quick T4 DNA Ligase in 2x Quick Ligation Reaction Buffer at 20°C for 15 minutes. The sequence of the NEBNext Adapter is listed as follows: 5′-/5Phos/GAT CGG AAG AGC ACA CGT CTG AAC TCC AGT CdUA CAC TCT TTC CCT ACA CGA CGC TCT TCC GAT C-s-T-3′. Then, 3 μl of USER Enzyme was added to the ligation mixture at 20°C for 15 minutes. Two rounds of nested anchored PCR, with primers complementary to the oligo tag, were used for target enrichment. Amplicons were purified by AMPure XP beads and sequenced on an Illumina sequencer. Restriction fragment-length polymorphism (RFLP) assays were used to assess oligo tag integration rates. Guide-seq v1.0.2 was used for GUIDE-seq analysis. The primer sequences for Guide-seq are listed in S2 Table.

**Mb4Cas12a-F370A-targetable sites in ClinVar and dbSNP databases.** Mutation sites in the human reference genome (GRCh38) were downloaded from the NCBI ClinVar and dbSNP databases. Mb4Cas12a-F370A-targetable sites and available gRNAs were analyzed according to the previously reported pipeline. Briefly, the flanking sequences (30 nucleotides from both upstream and downstream regions) of the mutation sites, including all mutations in dbSNP and "single-nucleotide variants (SNVs)" of "pathogenic" significance in ClinVar, were extracted from the human reference genome (GRCh38) for the subsequent analysis. Next, flanking regions were searched to find 5′PAM sequences (YYN) that could fit the SNV into the preferred editing window of Mb4Cas12a-F370A (positions 1 to 2, 4 to 5, 7, 10, 13 to 18 in the PAM proximal regions). At the same time, the corresponding gRNA spacer sequences could also be determined and output. The same procedure was used for the reverse complement strand. The resulting datasets were presented as S4 Table (targetable ClinVar and dbSNP variants with Mb4Cas12a-F370A).

**Quantification and statistical analysis.** All the data are shown as mean ± SD. Statistical analyses were conducted using GraphPad Prism 7. Student $t$ test or one-way analysis of variance (ANOVA) was used to determine statistical significance between 2 or more groups,

respectively. A value of $p < 0.05$ was considered to be statistically significant (* $p < 0.05$, ** $p < 0.01$, *** $p < 0.001$, **** $p < 0.0001$). All data needed to evaluate the conclusions in the paper are present in the paper and the Supporting inforation. All data needed to evaluate the conclusions in the paper are presented in S5 Table. The information on raw sequencing data is listed in S6 Table.

## Supporting information

**S1 Fig. The comparison of functional domains.** (**A**) Domain organization of LbCas12a [40]. (**B**) The percentages of protein identity between the newly identified Cas12a orthologs and LbCas12a.
(TIF)

**S2 Fig. Phylogenetic tree of selected Cas12a orthologs from different bacterial strains for activity screening.** Three validated Cas12a orthologs (blue colour) are used as references.
(TIF)

**S3 Fig. Direct repeat sequences of Cas12a.** (**A**) Direct repeat sequences of 12 Cas12a orthologs. LbCas12a direct repeat sequences are used as a reference. Sequences removed post crRNA maturation are indicated by a grey background; Stem sequences are indicated by a green background. Mature crRNA sequences are divided into 5 groups based on loop sequences. (**B**) The second structure of mature crRNA scaffolds.
(TIF)

**S4 Fig. The verification of Cas12a orthologs expression level.** (**A**) Protein expression levels of each Cas12a ortholog are analyzed by western blot. HEK293T cells without Cas12a transfection are used as a negative control (NC). (**B**) The efficient nuclear localization of each Cas12a ortholog is confirmed by immunofluorescence staining. The cells are stained with anti-HA antibodies. Scale bar: 10 μm.
(TIF)

**S5 Fig. Four Cas12a orthologs enable genome editing with YYN PAMs.** (**A**) Mb4Cas12a, BgCas12a, MoCas12a, and Ml2Cas12a enable genome editing with YYN PAMs in HEK293T cells. (**B**) Quantification of average editing efficiency for each Cas12a ortholog. ns stands for no significant. The numerical values underlying this figure can be found in S5 Table. The data represent the mean ± SD; $n = 3$. Two-way ANOVA, * $p < 0.05$. ** $p < 0.01$. ns means no significant.
(TIF)

**S6 Fig. Mb4Cas12a enables genome editing with YYN PAMs in multiple cell lines.** (**A**-**E**) Mb4Cas12a enables genome editing in HeLa (**A**), HCT116 (**B**), A375 (**C**), SH-SY5Y (**D**), and mouse N2a (**E**) cell lines. The numerical values underlying this figure can be found in S5 Table.
(TIF)

**S7 Fig. Test of Mb4Cas12a specificity with dinucleotide mismatches.** (**A**) The GFP-activation assay shows that Mb4Cas12a and LbCas12a are sensitive to the dinucleotide mismatches at crRNA positions 1–20. A schematic of the GFP-activation reporter is shown on the top; crRNAs are shown on the left; dinucleotide mismatches are indicated in red. The numerical values underlying this figure can be found in S5 Table. (**B**) Mb4Cas12a specificity is analyzed by the in vitro cleavage assay. Mb4Cas12a, crRNA, and DNA substrates are incubated at 37°C for 8 hours. crRNA sequences are shown below. Ctr, control, DNA substrates without

digestion. On, on-target crRNA.
(TIF)

**S8 Fig. Identification of residues that potentially forms hydrogen bonds.** (**A**) Analysis of the FnCas12a crystal structure showed that 8 residues form hydrogen bonds at the target DNA-crRNA interface. (**B**) The corresponding residues of Mb4Cas12a are identified by protein sequence alignment to FnCas12a. (**C**) The predicted F370 of Mb4Cas12a is structurally consistent with Y410 of FnCas12a by SWISS-MODEL.
(TIF)

**S9 Fig. Effects of 7 single mutations on Mb4Cas12a specificity.** (**A**) The GFP-activation assay was used to evaluate the effects of Mb4Cas12a. (**B**-**H**) The GFP-activation assay was used to evaluate the effects of mutations G257A (**B**), N272A (**C**), N276A (**D**), K291A (**E**), G305A (**F**), S307A (**G**), and K555A (**H**) on Mb4Cas12a specificity. The numerical values underlying this figure can be found in S5 Table.
(TIF)

**S10 Fig. Test of Mb4Cas12a and F370A specificity with GUIDE-seq.** (**A**-**H**) The genome-wide off-target effects of LbCas12a, Mb4Cas12a, and F370A are analyzed by GUIDE-seq. On-target and off-target sequences are shown on the left. Read numbers for on-target and off-target sites are shown on the right. Mismatches compared to the on-target site are shown and highlighted in colour.
(TIF)

**S11 Fig. Mb4Cas12a-F370A allows allele-specific genome editing in SH-SY5Y cells.** Mb4Cas12a-F370A allele specifically disrupts the single-nucleotide polymorphism (SNP) at locus rs28385702 (**A**), rs4927193 (**B**), rs747000387 (**C**), rs45545732 (**D**) in SH-SY5Y cells. PAMs are shown in light blue; target SNPs are shown in red; crRNAs are shown in purple; indel efficiencies are shown on the right. Data represent mean ± SD. $n = 3$. Student $t$ test, **** $p < 0.0001$. The numerical values underlying this figure can be found in S5 Table.
(TIF)

**S12 Fig. Mb4Cas12a-F370A allows allele-specific genome editing in C33A cells.** Mb4Cas12a-F370A allele specifically disrupts the single-nucleotide polymorphism (SNP) at locus rs28385702 (**A**), rs4927193 (**B**), rs747000387 (**C**), rs45545732 (**D**) in C33A cells. PAMs are shown in light blue; target SNPs are shown in red; crRNAs are shown in purple; indel efficiencies are shown on the right. Data represent mean ± SD. $n = 3$. Student $t$ test, ** $p < 0.01$, *** $p < 0.001$, **** $p < 0.0001$. The numerical values underlying this figure can be found in S5 Table.
(TIF)

**S1 Table. The human codon-optimized Cas12a sequences.**
(XLSX)

**S2 Table. Primers used in this study.**
(XLSX)

**S3 Table. Target sites used in this study.**
(XLSX)

**S4 Table. The crRNAs designed for clinically relevant variants in ClinVar and dbSNP variants with Mb4Cas12a-F370A.**
(XLSX)

**S5 Table. All underlying data in this study.**
(XLSX)

**S6 Table. The information on raw sequencing data in this study.**
(XLSX)

**S1 Raw Images. Raw images underlying data presented in this study.**
(PDF)

**S1 Data. Raw sequencing files for Sanger sequencing in this study.**
(7Z)

## Author Contributions

**Conceptualization:** Yongming Wang.

**Data curation:** Jingjing Wei, Ziwen Wang, Yuwen Tian.

**Formal analysis:** Jingtong Liu, Yuan Yang, Shengzhou Wang, Bao-Qing Gao.

**Funding acquisition:** Yongming Wang.

**Investigation:** Jingjing Wei.

**Project administration:** Yongming Wang.

**Supervision:** Song Gao, Li Yang, Junnan Tang, Yongming Wang.

**Writing – original draft:** Yongming Wang.

**Writing – review & editing:** Yongming Wang.

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
