## [Editor Report · Decision Letter 0]

1 Aug 2023

Dear Dr Wang, 

Thank you for submitting your manuscript entitled "Discovery and engineering of novel Cas12a nucleases for precision genome editing" for consideration as a Research Article by PLOS Biology.

Your manuscript has now been evaluated by the PLOS Biology editorial staff, as well as by an academic editor with relevant expertise, and I am writing to let you know that we would like to send your submission out for external peer review.

Once your full submission is complete, your paper will undergo a series of checks in preparation for peer review. After your manuscript has passed the checks it will be sent out for review. To provide the metadata for your submission, please Login to Editorial Manager (https://www.editorialmanager.com/pbiology) within two working days, i.e. by Aug 03 2023 11:59PM.

Kind regards,

Richard

Richard Hodge, PhD

rhodge@plos.org

PLOS

---

## [Decision Letter · Decision Letter 1]

6 Sep 2023

Dear Dr Wang,

Thank you for your patience while your manuscript " Discovery and engineering of novel Cas12a nucleases for precision genome editing " was peer-reviewed at PLOS Biology. Please accept my apologies for the delays that you have experienced during the peer review process. Your manuscript has now been evaluated by the PLOS Biology editors, an Academic Editor with relevant expertise, and by three independent reviewers. 

In light of the reviews, which you will find at the end of this email, we would like to invite you to revise the work to thoroughly address the reviewers' reports.

As you will see, the reviewers think the study is well conducted but raise overlapping concerns with the strength of the biochemical characterization and overstatements in regards to SNP specificity. Specifically, Reviewer #2 notes that additional evidence for SNP specificity should be provided in at least one cell line of each lineage and that 4 independent loci are included in the analysis. Reviewer #3 also notes that western blots should be provided for the orthologs tested in the GFP-activation assay to control for expression levels. 

Given the extent of revision needed, we cannot make a decision about publication until we have seen the revised manuscript and your response to the reviewers' comments. Your revised manuscript is likely to be sent for further evaluation by all or a subset of the reviewers.

**IMPORTANT - SUBMITTING YOUR REVISION**

*Re-submission Checklist*

*Published Peer Review*

*PLOS Data Policy*

*Blot and Gel Data Policy*

Sincerely,

Richard

Richard Hodge, PhD

rhodge@plos.org

REVIEWS:

Reviewer #1: The manuscript from W. J et al describes the discovery and engineering of novel Cas12a nucleases for precision genome editing. This work screened 14 Cas12a orthologs and characterized Mb4Cas12a in detail, and engineered Mb4Cas12a to improve specificity, enabling allele-specific editing of single-nucleotide polymorphisms (SNPs). The manuscript contains data on editing features in human cells and data on fidelity tests, as well as an incomplete biochemical characterization.

Overall, the manuscript has both strengths and weaknesses. Strengths include that new Cas12a systems were characterized, broadening our understanding of active Cas12a nucleases. Weakness is that the quality and reliability of the data need to be improved, and the manuscript has some flaws in writing.

Weaknesses:

1. In Fig 3A, the cleavage substrates of various PAMs belong to different target sites. It seems not convincing to compare PAM preference at different sites in vitro, due to the Cas nuclease may have site specificity, and the conclusions drawn from the cleavage assay are questionable.

2. Fig 6 shows the data of GUIDE-seq. the reads number of the Mb4Cas12a and mutant F370A is much less than that of LbCas12, and the difference in 4 of 6 sites (Fig 6A-C, 6F) even reach to 10-fold. Whether this obvious difference results from the editing activity at these target sites or from experimental manipulation needs plausible explanation.

3. The engineered Mb4Cas12a mutant F370 does not correspond to the sequence alignment in the SI figure. Fig S6B shows the 370 AA of Mb4Cas12a is Y.

4. The paper does not explain why alanine mutants of the amino acid which interact with DNA-crRNA duplex could improve Mb4Cas12a specificity.

5. The mutant of Mb4Cas12a had the ability to identify SNP sites and make selective editing, and the authors had a similar presentation: A highly specific CRISPR-Cas12j nuclease enables allele-specific genome editing (Science Advances). What would be the advantage of Cas12a reported in this manuscript over the previously reported Cas12j system?

6. The notes in Fig1C and Fig 4C mentioned Cas9, which seems a mistake in writting.

Reviewer #2: The manuscript submitted by Wei et al. titled "Discovery and engineering of novel Cas12a nucleases for precision genome editing" describes the characterization and functional analysis of several programmable endonucleases which are part of the Cas12a family. In this study, the authors focus on fourteen Cas12a orthologs and test their activity in vitro and in vivo using a variety of human cell lines. Wei and colleagues find that one of the orthologs, Mb4Cas12a, outperforms the other Cas12a variant tested with respect to on-target DNA cleaving activity and specificity (i.e. less off-target activity, more sensitive to mismatches). The latter motivates the authors to investigate whether the specificity and activity of Mb4Cas12a can be further improved by mutating eight highly conserved amino acid residues that 

potentially form hydrogen bonds at the target DNA-crRNA interface (based on previously published crystal structure data of FnCas12a). The authors show that the engineered mutant Mb4Cas12a-F370A outperforms both Mb4Cas12a and the reference LbCas12a in terms of specificity. In addition, the authors show to some extent that Mb4Cas12a-F370A is able to discriminate between alleles harbouring single nucleotide polymorphisms ("mismatches") and could potentially be further developed to treat autosomal dominant diseases 

through allele-specific disruption of mutant alleles.

Although the data presented in the manuscript is generally worked out well and with attention to detail, some figures require additional data to justify the scientific claims made by the authors.

These specific issues are described in more detail below.

Major issues:

- The authors claim that allele-specific editing is possible using an engineered version of Mb4Cas12a. However, allele-specific editing of the rs6684195 locus resulted in an indel rate of 28.3% at the target allele and 2.0% at the non-target allele. Although this is an almost 15-fold difference, one cannot claim that this method is specific. This difference is even smaller for the locus rs28385702 (indel rate: 5% at the target allele and 0.8% at the non-target allele). In addition, the authors provide data for only two genomic targets analysed for allele-specific editing in a single cell line (HEK293T). However, it remains unclear whether allele-specific editing is supported in other cell types as well. Together, this is insufficient evidence to claim that the new Cas12a variants enable allele-specific editing in human cell lines. Therefore, the authors need to either adjust their claim about Mb4Cas12a-mediated allele-specific editing or provide more supporting evidence showing at least similar fold differences in at least one cell line of each lineage (ectoderm, mesoderm, endoderm). In addition, the authors need to show that allele-specific editing is possible in at least four independent loci (ideally, randomly chosen from the 3084 clinically relevant variants listed in Table_S4). This is in line with previous work by Wang et al. - DOI: 10.1126/sciadv.abo6405 (2023) wherein allele-specific genome editing by Cas12j is described. 

Minor issues:

- Sanger sequencing and NGS data for any of the experiments described in the manuscript are not included and therefore need to be provided to be able to verify the results. 

- The methods section provides very little information for several key experiments or is missing altogether. For example, it is unclear how phylogenetic analysis was performed for Figure S1. Also, it is unclear how the GUIDE-Seq methodology was performed and the data analysed. Finally, it is unclear how the GFP reporter cell line used to assess PAM specificity was generated and characterised. Please provide details. 

- Figure 1 and 4 read "The cells without transfection of Cas9 were used as a negative control", but I assume the authors meant Cas12a.

- For Figures S7 and S8, please include data from WT Mb4Cas12a as a comparison. NB: I would suggest the authors combine these two figures into one.

- The data in Table S4 is shifted from row 441 onwards. Please modify.

- The names of the tables listed on the last page of the supplementals do not correspond with the files; Table_S3 is missing from the list and Table_S5 does not seem to exist (nor is it mentioned in the main manuscript).

Reviewer #3: The authors of this manuscript describe the characterization and application of new Cas12a nucleases. Using sequence comparisons, the authors start by identifying several new Cas12a proteins originating from various bacteria species. Using a GFP activation assay, the authors characterize the PAM sequence preference of all the tested variants. They then pursue characterization of both the efficiency and fidelity of the most promising variant: Mb4Cas12a (originating from Moraxella bovis). The authors subsequently generated mutated versions of Mb4Cas12a in order to improve its fidelity and validated that the mutations contribute towards this improvement.

Continuous characterization of Cas enzyme variants is crucial for the improvement of CRISPR/Cas based technologies. The various advancements that scientists are working on include increased efficiency, specificity, and flexibility (in the context of PAM preferences). The manuscript submitted by Wei et al. aligns with this overarching goal, as they explore new Cas12a variants, and therefore, it is timely, important, and has the potential for some impact. However, the novelty that the authors emphasize in this work about the SNP specificity of their new Cas12a variant is overstated as other Cas12a variants have been shown in literature to be highly specific and capable of discriminating SNPs. Aside from that, we believe the work presented in this manuscript is valuable to the scientific community and should eventually be published. However, we cannot recommend publication of the manuscript as it is currently constructed. Below we provide commentary, questions, and suggested experiments for the authors. These are intended to strengthen the authors' conclusions and focus the text to highlight novel and critical information. Finally, we note that it would be very helpful to include line numbers in any future iterations of this manuscript, as referring to specific text in their absence is tedious for reviewers. 

Major Points

1) Introduction: The authors mention: "A handful of Cas12a nucleases have been developed for genome editing, but none of them can discriminate single-nucleotide polymorphism (SNP)" as well as "Although Cas12a displays high specificity, they tolerate single or double mismatches". This claim seems overstated - for example, AsCas12a Ultra (PMID 34162850) shows excellent avoidance of off-target cutting (we recognize that this is not exactly the same as a demonstrated ability to distinguish SNPs). Additionally, there are numerous examples of Cas12a losing nearly all activity on the basis of a single point mutant. As we noted above, we don't think SNP detection is an important deliverable of this manuscript, so we highly encourage the authors not to overplay the novelty of their claims in this area. 

2) Results - Screen of active Cas12a orthologs by the GFP-activation assay: It appears that the sequence identity percentages between the various Cas12a orthologs and LbCas12a in Table 1 are relatively low. A sequence comparison of the functional domains could be more informative.

3) Results - Screen of active Cas12a orthologs by the GFP-activation assay: After transfection of the various Cas12a variants in the cells, the authors measure GFP expression as a readout for editing efficiency (Fig1C). To rule out that the difference in performance is not due to differences in levels of expression the authors should perform western blots for all the orthologs tested. The authors should also address the efficient nuclear localization of these Cas12a variants by staining or by performing nuclei specific western blot.

4) Results - Analysis of Mb4Cas12a specificity: To increase Mb4Cas12a specificity, the authors identify critical amino acid residues within the protein to mutate. They rely on the crystal structure of FnCas12a and compa

---

## [Decision Letter · Decision Letter 2]

6 Feb 2024

Dear Dr Wang,

Thank you for your patience while we considered your revised manuscript "Discovery and engineering of novel Cas12a nucleases for precision genome editing" for consideration as a Research Article at PLOS Biology. Please accept my apologies for the delays that you have experienced during this round of the peer review process. Your revised study has now been evaluated by the PLOS Biology editors, the Academic Editor and the original reviewers. 

As you will see, the reviewers are generally satisfied with the revision and the additional data included. However, Reviewer #2 notes a contradiction in the GUIDE-seq data to assess off-target effects of Mb4Cas12a, since it is claimed that that the engineered Mb4Cas12a-F370A is more specific than its wild-type counterpart, but this does not seem to be shown by the sequencing data. We ask that a further explanation/clarification or additional experimentation is provided to support these claims.

In light of the reviews, which you will find at the end of this email, we are pleased to offer you the opportunity to address the remaining points from Reviewer #2 in a revision that we anticipate should not take you very long. We will then assess your revised manuscript and your response to the reviewers' comments with our Academic Editor aiming to avoid further rounds of peer-review, although might need to consult with the reviewers, depending on the nature of the revisions.

In addition, I would be grateful if you could please make sure to address the following data and other policy-related requests that I have provided below (A-F):

(A)We would like to suggest the following modification to the title: 

““Engineering of a high-fidelity Cas12a nuclease variant capable of allele-specific editing”

(B) Please deposit the GUIDE-seq sequencing data (Figure 6, S11) and the deep sequencing data (Figure 2) in a public repository such as the GEO. Please ensure that the data is made publicly available and provide the accession number of the deposition in the Data Availability Statement in the online submission form.

(C) Please also ensure that each of the relevant figure legends in your manuscript include information on *WHERE THE UNDERLYING DATA CAN BE FOUND*, and ensure your supplemental data file/s has a legend.

(D) We require the original, uncropped and minimally adjusted images supporting all blot and gel results reported in the following Figures:

Figure 3A-B, 4B, S4A, S7B

We will require these files before a manuscript can be accepted so please prepare and upload them now. Please carefully read our guidelines for how to prepare and upload this data: https://journals.plos.org/plosbiology/s/figures#loc-blot-and-gel-reporting-requirements

(E) Please ensure that your Data Statement in the submission system accurately describes where your data can be found and is in final format, as it will be published as written there. This includes referencing where the sequencing data can be found in the GEO.

(F) Please note that per journal policy, the specific species where the Cas12a nuclease comes from should be clearly stated in the abstract of your manuscript.

**IMPORTANT - SUBMITTING YOUR REVISION**

*Resubmission Checklist*

*Published Peer Review*

*PLOS Data Policy*

*Blot and Gel Data Policy*

Sincerely,

Richard

Richard Hodge, PhD

rhodge@plos.org

REVIEWS:

Reviewer #1: The authors have fully addressed my previous concerns.

Reviewer #2: The revised manuscript shows additional off-targets for POLQ (11 reads), which were absent in the original manuscript. The authors claim that the engineered Mb4Cas12a-F370A is more specific than its wild-type counterpart, but this is contradicted by the GUIDE-seq data presented in Figure 6. In fact, Mb4Cas12a-F370A seems far less specific than Mb4Cas12a based on the (proportionally) higher number of reads for POLQ off-targets 1, 2 and 3. Interestingly, these off-targets contain up to three mismatches but are still efficiently cleaved by Mb4Cas12a-F370A. The manuscript states: "LbCas12a generated 28 off-targets across six tested loci. In contrast, Mb4Cas12a generated six off-targets across six tested loci, while F370A generated seven off-targets across six tested loci.". However, no explanation is provided for this observation.

Reviewer #3: The authors have sufficiently responded to prior comments, and this manuscript will be a nice addition to the field.

---

## [Editor Report · Decision Letter 3]

25 Apr 2024

Dear Dr Wang,

Thank you for your patience while we considered your revised manuscript "Discovery and engineering of novel Cas12a nucleases for precision genome editing" for publication as a Research Article at PLOS Biology. This revised version of your manuscript has been evaluated by the PLOS Biology editors and the Academic Editor.

Based on our Academic Editor's assessment of your revision, I am pleased to say that we are likely to accept this manuscript for publication, provided you satisfactorily address the following data and other policy-related requests that I have provided below:

(A) We would like to suggest the following modification to the title:

“Engineering of a high-fidelity Cas12a nuclease variant capable of allele-specific editing”

(B) In your conflict of interest statement in the online submission form, please provide additional details about the specific authors who have applied for the patent and any patent numbers which have been provided. 

(C) You may be aware of the PLOS Data Policy, which requires that all data be made available without restriction: http://journals.plos.org/plosbiology/s/data-availability. For more information, please also see this editorial: http://dx.doi.org/10.1371/journal.pbio.1001797

Thank you for already providing the raw data underlying the figures in the Table S5 file. However, we note that the raw data for Figure 5B is currently missing in the supplementary file?

(D) Thank you for already providing the raw sequencing data in the Table S6 file. However, I note that the raw sequencing files for the GUIDE-seq data (Figure 6, S11), deep sequencing (Figure 2) and Sanger sequencing (Figure 3C) appears to be missing from this file. Given the potential size of the sequencing files, I would be grateful if you could please deposit this data in the GEO data repository. Please ensure that the data is made publicly available and provide the accession number in the Data Availability Statement in the online submission form. 

(E) Please also ensure that each of the relevant figure legends in your manuscript include information on *WHERE THE UNDERLYING DATA CAN BE FOUND*, and ensure your supplemental data file/s has a legend.

(F) Please ensure that your Data Statement in the submission system accurately describes where your data can be found and is in final format, as it will be published as written there. 

(G) Per journal policy, if you have generated any custom code during the curse of this investigation, please make it available without restrictions upon publication. Please ensure that the code is sufficiently well documented and reusable, and that your Data Statement in the Editorial Manager submission system accurately describes where your code can be found. 

(H) We require the original, uncropped and minimally adjusted images supporting all blot and gel results reported in the following Figures:

Figure 3A-B, 4B, S4A, S7B

We will require these files before a manuscript can be accepted so please prepare and upload them now. Please carefully read our guidelines for how to prepare and upload this data: https://journals.plos.org/plosbiology/s/figures#loc-blot-and-gel-reporting-requirements

(I) Please note that per journal policy, the specific species where the Cas12a nuclease was obtained from should be clearly stated in the abstract of your manuscript. 

We expect to receive your revised manuscript within two weeks. 

*Published Peer Review History*

*Press*

Sincerely,

Richard

Richard Hodge, PhD

rhodge@plos.org

PLOS

---

## [Editor Report · Decision Letter 4]

16 May 2024

Dear Yongming,

On behalf of my colleagues and the Academic Editor, Jacob Corn, I am pleased to say that we can accept your manuscript for publication, provided you address any remaining formatting and reporting issues. These will be detailed in an email you should receive within 2-3 business days from our colleagues in the journal operations team; no action is required from you until then. Please note that we will not be able to formally accept your manuscript and schedule it for publication until you have completed any requested changes.

PRESS

Best wishes, 

Richard

Richard Hodge, PhD

rhodge@plos.org

PLOS
